# Decentralized, Decomposition-Based Observation Scheduling for a Large-Scale Satellite Constellation

**Primary Keywords:** *(1) Applications; (7) Multi-Agent Planning;*

## Abstract

Deploying multi-satellite constellations for Earth observation requires coordinating potentially hundreds or thousands of spacecraft. With increasing on-board capability for autonomy, we can view the constellation as a multi-agent system (MAS) and employ decentralized scheduling solutions. We formulate the problem as a distributed constraint optimization problem (DCOP) and desire scalable inter-agent communication. The problem consists of millions of variables which, coupled with the structure, make existing DCOP algorithms inadequate for this application. We develop a scheduling approach that employs a well-coordinated heuristic, referred to as the Geometric Neighborhood Decomposition (GND) heuristic, to decompose the global DCOP into sub-problems as to enable the application of DCOP algorithms. We present the Neighborhood Stochastic Search (NSS) algorithm, a decentralized algorithm to effectively solve the multi-satellite constellation observation scheduling problem using decomposition. In full, we identify the roadblocks of deploying DCOP solvers to a large-scale, real-world problem, propose a decomposition-based scheduling approach that is effective at tackling large scale DCOPs, empirically evaluate the approach against other baselines to demonstrate the effectiveness, and discuss the generality of the approach.

## Introduction

Large-scale, Earth observing satellite constellations with hundreds of spacecraft are becoming increasingly prominent in order to monitor Earth phenomena. Spire, Satellogic, Canon, SatRev, Spacety, Planet Lab's Dove, and SkySat are several examples (NewSpace 2023). Observation scheduling for a large-scale constellation requires fusing information from many sources and tasking space assets that have varying constraints, capabilities, and visibility of Earth targets. In addition to Earth observation, satellites are deployed for a variety of applications, including forming large internet constellations (SpaceX 2023). Any multi-satellite constellation that requires coordinating agents poses a challenging planning and scheduling problem.

In practice, satellite observation scheduling is typically done in a centralized fashion, where a single controller develops a single schedule that specifies the actions of every satellite (Shah et al. 2019). Even most non-operational technology efforts are centralized (Boerkoel et al. 2021; Nag, Li, and Merrick 2018). While centralized approaches can provide high-quality solutions, reliance on a single computing source makes the approaches vulnerable to single point failures and can increase communications burden.

Framing the constellation as a multi-agent system (MAS) enables the application of decentralized scheduling solutions. Decentralized scheduling addresses both the system's robustness and the vulnerabilities of a central controller (Bonnet and Tessier 2008; Phillips and Parra 2021). Many MAS problems are framed to optimize a global cost function where individual agents control the parameters of the function. Typically, the agents in the system communicate to coordinate their parameter assignments.

In many applications, communication may be unreliable for large message volume. For example, orbiters around a comet or the sun may experience communication interference or infrequent line of sight with each other. For an Earth orbiting constellation, satellites may have limited cross-link capability. For this reason, we desire algorithms that procure a limited amount of messaging. Formulating the problem as a distributed constraint optimization problem (DCOP), we aim to produce high quality scheduling solutions.

The problem requires agents to coordinate the assignments of millions of variables and the desire for limited communication make the direct application of DCOP algorithms inadequate. However, by decomposing the global problem, we can deploy DCOP algorithms to solve smaller sub-problems. We construct a heuristic, called the *Geometric Neighborhood Decomposition heuristic* (GND), that partitions the agents and requests in a coordinated fashion as to instantiate sub-problems that are advantageous to solve. Each agent individually computes the heuristic using only knowledge of the requests to schedule and the configuration of the constellation. The heuristic is grounded in geometric computation and is composed of three layers that partition the agents and requests into sub-problems. The goal is for the constellation to maximize the number of requests satisfied, while adhering to downlinks and memory constraints.

The heuristic is parameterized such that the sub-problems produced can be of arbitrary size. Through this decomposition, we can deploy DCOP algorithms on constant sized sub-problems rather than the global problem that scales with the number of agents and requests. To solve each sub-problem using communication, we build on the *Broadcast Decentralized* algorithm (BD)

(Parjan and Chien 2023), an adaptation of two incomplete DCOP algorithms, *Maximum Gain Messaging* (MGM) and *Distributed Stochastic Search Algorithm* (DSA) (Zhang et al. 2005), for the application of multi-satellite observation scheduling. We refer to our developed algorithm as *Neighborhood Stochastic Search* (NSS). Our algorithm extends the BD algorithms in two major aspects: (1) it is scalable to large problem instances in both computational complexity and communication complexity, and (2) it enables constraint reasoning, such as resource constrained scheduling.

Empirical results demonstrate the efficacy of our approach on small and large problem instances compared to decentralized and centralized baselines. On small problem instances, we show the gap to optimal solutions, while large problem instances enforce the performance at scale, including run-time results. We aim to close the gap between decentralized solutions and centralized solutions while precluding an infeasible amount of messaging.

Our contributions in this paper are hence:

1. uncovering the obstacles in applying existing DCOP techniques to the large-scale, multi-satellite decentralized scheduling problem,

2. introducing a decomposition-based heuristic approach to solve the scheduling problem and presenting the NSS algorithm, and

3. demonstrating the efficacy of our approach on realistic problem instances.

## Related Work

There are many aspects of satellite observation planning ranging from visibility computation, to downlink scheduling, to constraint based task allocation. This paper is mostly concerned with the last subject. Previous work has predominantly focused on centralized solutions to the multi-satellite, resource-constrained scheduling problem (Augenstein et al. 2016; Nag, Li, and Merrick 2018; Shah et al. 2019; Squillaci, Roussel, and Pralet 2021; Boerkoel et al. 2021; Squillaci, Pralet, and Roussel 2023; He et al. 2018; Eddy and Kochenderfer 2021; Globus et al. 2004). More recently, decentralized scheduling approaches have proposed auction-based methods (Picard 2021; Phillips and Parra 2021) and heuristic search based methods relying on broadcasting (Parjan and Chien 2023). The auction-based methods rely on a centralized controller to act as an auctioneer and has a prohibitive communication and computational complexity. Each agent exchanges a polynomial (in the requests) number of messages with the auctioneer. Removing the central auctioneer is possible, but results in an explosion of the communication complexity depending on the network topology.

We build on the approach presented by Parjan and Chien (2023), which attempts to address some of the limitations of the auction-based methods. In their approach, referred to as BD, each agent uses globally communicated satisfaction information as a search heuristic. The main limitation is that this approach requires each agent to send a high volume of messages to every other agent, resulting in a communication complexity at each iteration that is polynomial in the number of agents and requests.

Multi-satellite observation scheduling has been framed as a DCOP previously (Picard 2021; Parjan and Chien 2023). DCOP algorithms tend to suffer from significant computational complexity or a large reliance on communication. Solving a DCOP optimally is known to be NP-Hard (Modi et al. 2005). Complete algorithms, such as *SyncBB* (Hirayama and Yokoo 1997), *ADOPT* (Modi et al. 2005), or *OptAPO* (Mailler and Lesser 2004) are computationally infeasible for our problem scale.

On the other hand, incomplete DCOP algorithms, such as *Max-Sum* (Stranders et al. 2009), *Maximum-Gain Messaging* (MGM) (Maheswaran et al. 2004), *Distributed Stochastic Search* (DSA) (Zhang et al. 2005), or *Distributed Gibbs* (D-Gibbs) (Nguyen, Yeoh, and Lau 2013), which trade off optimality for scalability, require notable messaging. MGM and DSA are two search algorithms that are the foundation of BD, and hence NSS. MGM and DSA perform local search to iteratively improve the global solution. Genetic algorithms have also been applied, incurring similar complexity as MGM and DSA (Mahmud et al. 2019).

NSS is motivated by *Region-optimal* algorithms (Pearce and Tambe 2007). These algorithms solve sub-problems optimally, reducing the cost of complete algorithms, albeit to the size of sub-problems. NSS extends *Region-optimal* algorithms by obtaining incomplete solutions to sub-problems, reducing the complexity when sub-problems remain large. Another divide-and-conquer method for solving DCOPs is the application of the distributed large neighborhood search (DLNS) framework to DCOPs (Hoang et al. 2018).

In the next sections, we discuss the challenges of applying these algorithms to the multi-satellite constellation observation scheduling problem, and show how heuristically decomposing the problem can overcome the hurdles in their deployment while still providing high quality solutions.

## Problem Formulation

In this section, we outline the *multi-satellite constellation observation scheduling problem* (COSP). The main application of COSP is for Earth observation, however the formulation extends to orbits around other bodies. We start by formally defining the problem. Then, we present the problem as a DCOP and discuss theoretical properties of the problem, including the barriers to applying current DCOP algorithms.

### Defining COSP

The components of COSP are defined below.

1. $H = [h_s, h_e]$: the scheduling horizon.

2. $\mathcal{K}$ : the set of orbital planes. An orbital plane defines the geometric plane that contains a collection of satellite orbits. We denote $K \in \mathcal{K}$ as an orbital plane, and $k \in K$ for the specific orbit of a satellite within that plane. Figure 1 shows the orbits defined by a single orbital plane with 5 satellites.

3. $A$: the set of agents. Each $a_i \in A$ is a satellite in the constellation. We define $a_i = (k, m)$ where $k$ is the orbit of the satellite and $m \in \mathbb{R}^+$ is the memory capacity. The notation $k(a_i)$ and $m(a_i)$ denote these values for agent $a_i$, and will use the same notation for equivalent indexing.

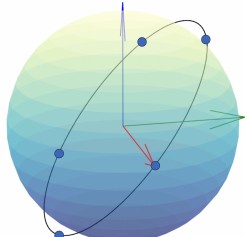

Figure 1: Visualization of an orbital plane with 5 satellites.

4. $T$: the set of point targets on Earth. Each $t_i \in T$ is defined as $t_i = $ (*lat*, *lon*).

5. $R$: the set of requests. Each $r_i \in R$ is defined as $r_i = (t, h)$ which denotes the target to observe, $t \in T$, and when to observe, $h \subset H$.

6. For each agent, we define the following sets.

$S_{a_i}$: the set of possible *request fulfillments* for agent $a_i$. A request fulfillment, $s_j \in S_{a_i}$, is a task that can be scheduled to satisfy a request. We define $s_j = (r, h, m)$ where $r \in R$ is the request being satisfied, $h \subset h(r)$ is the interval of the observation (including processing and slewing time), and $m \in \mathbb{R}^+$ is the amount of memory required.

$\mathcal{X}_{a_i}$: the set of Boolean decision variables for agent $a_i$. For each $s_j \in S_{a_i}$ we define the Boolean decision variable $x_j \in \mathcal{X}_{a_i}$ where $x_j = 1 \iff$ agent $a_i$ schedules task $s_j$. We denote $x(s_j) = x_j$ for agent $a_i$.

$D_{a_i}$: the set of downlinks for agent $a_i$. A downlink, $d_j \in D_{a_i}$, is defined as $d_j = (h, m)$ where $m \in \mathbb{R}^+$ is the maximum amount of data that can be downlinked, and the interval $h \subset H$ is the time window for the downlink. No possible tasks occur during a downlink, and all downlinks are mandatory.

$C_{a_i} = C_{D_{a_i}} \cup C_{S_{a_i}}$: the set of constraints where

$$C_{D_{a_i}} = \bigcup_{d_j \in D_{a_i}} c_{d_j}, \text{ and}$$

$$c_{d_j} = \sum_{s_l \in S_{a_i}^{d_j}} x(s_l) \cdot m(s_l) \leqslant \text{MIN}(m(a_i), m(d_j)).$$

Here, $S_{a_i}^{d_j}$ denotes the set of possible tasks for which the soonest downlink window in the future is $d_j$. This constraint enforces that agent $a_i$ never exceeds its memory capacity and all taken observations can be downlinked at the soonest opportunity. We define

$$C_{S_{a_i}} = \bigcup_{s_j, s_l \in S_{a_i}} c_{s_j, s_l}, \text{ where}$$

$$c_{s_j, s_l} = \left[ x(s_j) \cdot x(s_l) + \mathbb{I}(h(s_j) \cap h(s_l) \neq \varnothing) \leqslant 1 \right].$$

This constraint ensures that no tasks are scheduled to overlap. Here, $\mathbb{I}$ denotes the indicator function.

The goal of the optimization problem is to maximize the number of requests satisfied while not violating the constraints of any agent. A solution, $X$, is the assignment of each $x \in \mathcal{X}_{a_i}$ such that $C_{a_i}$ is satisfied for all $a_i$.

## Formulating COSP as a DCOP

We can formulate the above problem structure as a distributed constraint optimization problem (DCOP), similar to previous work (Picard 2021; Parjan and Chien 2023). The DCOP is a five-tuple $\langle A, \mathcal{X}, \mathcal{D}, \mathcal{F}, \alpha \rangle$ which we define for our problem below.

- $A$: the set of satellites as previously defined.
- $\mathcal{X} = \bigcup_{a_i \in A} \mathcal{X}_{a_i}$: the set of Boolean decision variables for every agent's possible task set as previously defined.
- $\mathcal{D} = \bigcup_{x \in \mathcal{X}} \{0, 1\}$: all variable domains are Boolean.
- $\mathcal{F}(X) = \bigcup_{a_i \in A} f_{a_i}(X_{a_i}) \cup \bigcup_{r_j \in R} f_{r_j}(X_{r_j})$ where

$$f_{a_i}(X_{a_i}) = \begin{cases} 0 & C_{a_i} \text{ satisfied by } X_{a_i} \\ -\infty & \text{else} \end{cases}$$

and

$$f_{r_j}(X_{r_j}) = 1 - \prod_{x \in X_{r_j}} (1 - x).$$

Here, $X_{a_i}$ is the set of variables in the solution, $X$, such that $\alpha(x) = a_i$, and $X_{r_j}$ is the set of variables such that $x = x(s_l)$ and $r(s_l) = r_j$. See that $f_{r_j}(X_{r_j}) = 1$ iff there exists a satellite satisfying request $r_j$ and $f_{a_i}(X_{a_i}) = 0$ iff agent $a_i$ has a schedule that satisfies its constraints.

- $\alpha(x) = a_i \iff x \in \mathcal{X}_{a_i}$ maps a variable to the agent that can schedule the associating request fulfillment.

The goal of a DCOP is to obtain an assignment of all variables as to maximize (or minimize) the sum of the utility functions, $X^* = \text{argmax}_X \sum_{f \in \mathcal{F}} f(X^S)$.

While the problem can be represented as a DCOP, there exist roadblocks to applying existing DCOP algorithms. The constraint graph defined by this formulation has a minimum of $|A| + |R| \approx 10^3$ complete sub-graphs derived from $f_{a_i}, f_{r_j}$. Each $a_i \in A$ contributes a clique of size $|\mathcal{X}_{a_i}|$, and each $r_j \in R$ contributes a clique of size $|\mathcal{X}_{r_j}|$. In many problem instances, request durations are long enough such that the majority of agents are able to satisfy any particular request. This results in $|\mathcal{X}_{a_i}| = \Omega(|R|)$ and $|\mathcal{X}_{r_j}| = \Omega(|A|)$. In addition, most of these cliques are highly connected to each other, resulting in a cyclic graph. Figure 2 shows an example of the structure we discuss for a problem instance with 3 agents and 4 requests.

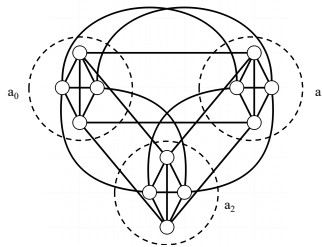

Figure 2: Constraint graph with $|A| = 3$ and $|R| = 4$. Nodes denote the variables, and edges denote a shared constraint.

Recall that we consider constellations with hundreds of agents and thousands of requests. Therefore, since each

agent $a_i$ controls all variables in $\mathcal{X}_{a_i}$, the neighborhood of variables for an *agent* is $\Omega(|A| \cdot |R|) \approx 10^5$ variables. The problem scale coupled with the structure make it unsuitable for many DCOP algorithms. Complete DCOP algorithms are simply infeasible for the problem size, having a computational complexity of $O(2^{|A| \cdot |R|}) \approx 10^{30102}$. Incomplete DCOP algorithms, such as *Max-Sum* (Stranders et al. 2009) or *Maximum Gain Messaging* (MGM) (Maheswaran et al. 2004), require many iterations, where at every iteration, each agent would exchange $\Omega(|A|)$ messages, each with size $\Omega(|R|)$, which is on the order of $10^7$ total volume if agents send direct messages to each other (Fioretto, Pontelli, and Yeoh 2018). Finally, we mention that the DLNS and region-optimal approaches, require defining sub-problems. One application of our approach is outlining methods for sub-problem selection. However, the above two algorithms still incur substantial costs when sub-problems remain large.

We mention one distinction between the scheduling problem we will examine in this paper and a traditional DCOP. In standard DCOPs, agents know the variables and constraints of neighboring agents (Fioretto, Pontelli, and Yeoh 2018). An agent is unaware of the *values* of the variables that other agents control, but is privy to their existence. In our multi-satellite constellation, we only assume that agents are aware of the existence of other *agents*, but have no knowledge of the request fulfillments (variables) other agents are attempting to schedule. The implication is that an agent does not know the full structures of the utility functions for which its variables are affiliated. Note, it takes one broadcast for each agent to share their request fulfillments before the problem becomes a standard DCOP as outlined in the previous section. This small nuance is motivated by the requirements of the application, but does not impact the approach significantly from applying to other problems.

## Heuristic Decomposition

In this section, we outline the construction of the *Geometric Neighborhood Decomposition heuristic* (GND) that decomposes the global problem. GND, which is computed individually by each agent, inherently coordinates agents without communication. This heuristic is grounded in geometry, and is composed of three layers: (1) the global supply layer, (2) the inter-neighborhood delegation layer, (3) and the intra-neighborhood delegation layer.

The first layer addresses the nuance of the scheduling problem mentioned in the previous section, while the latter two layers act to partition the agents and requests into sub-problems. An agent computes the heuristic values only relevant to itself, remaining unaware of the heuristic computation of other agents.

## Global Supply

In our application, *supply*, or the number of agents capable of satisfying a request, is important as there are observations for which only a few satellites have visibility, as well as requests that the entire constellation can service. Identifying supply enables agents to make informed decisions as to minimize redundant observations and collectively service more

requests. The lack of knowledge of other agent variables results in agents being unaware of the global supply.

In the traditional DCOP, the supply is trivial to compute. For a request, $r$, the supply is the number of agents that have a variable $x = x(s_j)$ such that $s_j = (r, h, m)$, which is known since these agents all share a constraint.

We use the geometry of the satellite orbits to estimate the visibility of a ground target for each orbital plane, obtaining an approximation of the supply. Specifically, the supply of a request, $r$, provided by an orbital plane, $K$, is determined as the duration of time that the request is in the *longitudinal cross-tracks* of the plane times the *number of agents that pass over a point per epoch*. The longitudinal cross-tracks are the intervals in which the target is within a visible range of an orbital plane. This is determined by the point of closest approach, satellite slewing capabilities, and field of view.

Figure 3 illustrates the geometric interval that a ground target is in the cross-tracks. In the figure, $Z_{ECI}$ denotes the rotation axis of Earth. $P_1$ and $P_2$ are the bounding planes of the cross-tracks either side of the orbital plane. The green area then depicts the region for which any ground target might be within visibility of a satellite in the orbital plane. The right ascension bounds depict the interval of potential visibility for a specific ground target. Combining that interval with the number of agents that pass over a point per epoch, we obtain the estimate of supply. The latter piece of information is determined by the fixed amount of time it takes any satellite in the orbital plane to complete one full orbit and the number of agents in the plane.

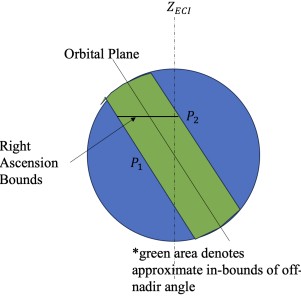

Figure 3: The right ascension bounds show the interval a target is within potential visibility of the orbital plane.

Whether the supply is estimated or known exactly makes little difference to the approach. However, in this application, knowing the supply is an unrealistic assumption.

## Inter-Neighborhood Delegation

The supply layer provides a global view of the problem to each satellite. Considering the number of satellites that can service an observation request puts the contention into perspective. The inter-neighborhood heuristic delegates a request to a fixed subset of agents.

Each orbital plane, $K \in \mathcal{K}$, provides a natural grouping of agents into a neighborhood. Satellites in the same orbital plane experience similar relative geometries of Earth targets, making their view of the problem homogeneous.

They typically share the ability to satisfy the same requests, making orbital planes a logical choice of neighborhoods. We acknowledge that this neighborhood selection exploits specifics of our domain, and that neighborhood selection in other domains is not always as clear-cut.

In this section, and future sections, we use the term orbital plane and neighborhood interchangeably. It is important to note that our use of the term neighborhood is different from the typical notion of a neighborhood in DCOPs. In a DCOP, the neighborhood refers to agents that share a constraint. Agents in an orbital plane typically share many constraints, however, they also share constraints with agents in other orbital planes. At an abstract level, this layer of the heuristic delegates control of each utility function, $f_{r_j}$, to a partition of agents.

We define the inter-neighborhood delegation heuristic based on properties of an orbital plane and the request. For a given request, $r$, the neighborhood delegated to it is determined as the orbital plane with the most agents for which there is non-zero supply. Ties are broken by selecting the plane with the minimum distance to the request target, $t(r)$, over the request window. The minimum distance is estimated using three points over the continuous interval of the request horizon, $h(r)$: the start, the end, and the midpoint. This tie-breaking can be viewed as a unique secondary estimate of the supply.

The inter-neighborhood delegation serves as a complete partitioning of the global problem into sub-problems. For each request, $r$, there exists exactly one plane, $K$, such that the request will be delegated. Consider that agent $a_i$ disregards all variables, $x$, for which the associated request is not delegated to the neighborhood of $a_i$. We can then remove the factors containing $x$ from each $f_{r_j}$ and $f_{a_i}$. The problem has now been decomposed such that an agent is only neighbors (by the DCOP notion) with other agents in the same neighborhood. From this we obtain a partitioning into $|\mathcal{K}|$ sub-problems in which the agents of each sub-problem are the agents in an orbital plane, $K$, and the variables are determined by the inter-neighborhood delegation heuristic.

Later, we will present the constellation, but as a consideration we note here that $|\mathcal{K}| = 4$ in our evaluations. Therefore, while this partitioning is substantial, it is not necessarily sufficient to reduce the problem scale to a desired level.

### Intra-Neighborhood Delegation

The value of the supply heuristic and the inter-neighborhood heuristic are not unique for agents in the same orbital plane. The final layer in the decomposition further partitions neighborhoods into smaller problems. The intra-neighborhood heuristic is driven by agent biases, where agents with the same biases form a subset of the partition.

First, we define the bias of an agent, denoted $b$. A bias is parameterized by a periodicity, $\rho \in \mathbb{N}$. The periodicity both determines the number of unique biases and the number of agents for which the bias repeats in a neighborhood. To compute the bias, we index each agent in an orbital plane, $K$, by arbitrarily selecting an agent 0 and ordering all the agents in the plane from 0 to $|K| - 1$ by moving counter-clockwise. The bias, $b \in [0, \rho)$, is computed as an integer

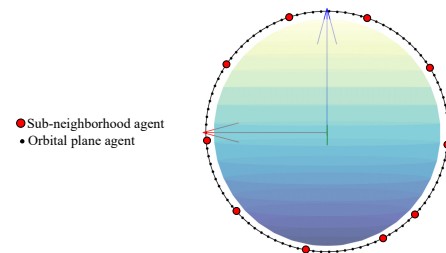

Figure 4: Partition within an orbital plane with $\rho = 10$.

based on the index, $i$, of the agent: $b \equiv i \mod \rho$.

The periodicity of the bias means all agents indexed $i + n \cdot \rho$ possess the same bias for all $n \in \mathbb{N}$. Increasing the periodicity creates more sub-groups of agents with the same bias. The motivation for spacing agents with the same bias as opposed to selecting consecutive agents is to diversify the sub-neighborhood's collective visibilities and geometries. This creates more advantageous sub-problems to solve by enabling a wider birth of opportunities for observation. The bias for a single agent is fixed, therefore we assume it is known by all agents apriori with the configuration of the satellite constellation.

We consider two biases. The supply bias sub-divides the original supply thresholds into $\rho$ partitions, and an agent with bias, $b$, will be biased towards requests that fall into the sub-bucket indexed $b$. The request target position bias examines the latitude and longitude of the ground target, $t(r)$. For both coordinates, we add a bias if the degree times ten modular $\rho$ is equivalent to the agent bias $b$. We multiply by ten to give more precision to the bias. Targets tend to be clustered in small geographic regions, so using more precision provides more diversity in this bias. We mentioned previously that many agents experience similar relative geometry with targets in terms of time and space. The latitude and longitude bias aims to disrupt that homogeneity across agents. A request is delegated to the sub-neighborhood of agents for which the shared bias has the highest value.

The intra-neighborhood delegation partitions the agents in an orbital plane into $\rho$ sets, where a set is defined by agents with the same bias. Tuning $\rho$ enables us to create partitions of arbitrary size, but at a potential reduction of coordination among agents.

### Complete Heuristic

The GND heuristic, $\Upsilon_{a_i} : R \rightarrow \{0, 1\}$, is a function computed by an agent that maps a request fulfillment to a Boolean value. The heuristic identifies the subset of requests assigned to the partition containing an agent. Iff $\Upsilon_{a_i}(r) = 1$, then the request $r$ is within agent $a_i$'s partition. The agent's within the partition are fixed based on the geometry of the constellation and the parameter $\rho$. We tuned $\rho$ and take the value $\rho = 5$. We mention that small deviations in $\rho$ had minimal effect on the overall performance, but drastic changes did worse. Taking $\rho = 5$ results in all sub-problem having less than 20 agents, reducing the size from the global problem by an order of magnitude.

# Scheduling Solutions to COSP

In this section, we present the algorithms we evaluate, including *Neighborhood Stochastic Search* (NSS).

## Fully Decentralized

By fully decentralized solutions, we refer to decentralized algorithms that do not rely on inter-agent communication. We present three baseline algorithms.

1. *Random*. Each agent shuffles its set of request fulfillments, $S_{a_i}$. The shuffled request fulfillments are iterated through and scheduled if they do not violate constraints.

2. *Greedy Start Time*. Each agent sorts its request fulfillments based on increasing start time. The sorted request fulfillments are iterated through and scheduled if they do not violate constraints.

3. *Portfolio Greedy*. Each agent samples a heuristic from the portfolio of heuristics, $\Pi$, uniformly at random. The heuristic defines the greedy insertion order into the schedule. The portfolio consists of four heuristics: random, start time, memory usage, and off-nadir angular separation. The random heuristic assigns a random value to each request fulfillment. We note that each heuristic was evaluated individually, and the portfolio does not contain a dominating heuristic.

In addition, to demonstrate the effectiveness of the partitioning, we also evaluate the *Greedy Start Time* algorithm, but each agent first computes the decomposition using GND and only considers the partitioned requests. We call this fully decentralized approach *Decomposition Heuristic*.

## Centralized Algorithm

The centralized algorithm we employ is an adaptation of *Squeaky Wheel Optimization* (SWO) (Joslin and Clements 1999). SWO is an incomplete centralized search algorithm. Over the course of iterations, SWO heuristically creates schedules based on priorities assigned over previous iterations. In our implementation, SWO sorts the requests based on their priority, breaking ties with the exact supply. It then randomly selects an available satellite to schedule the request. Requests that are not scheduled on previous iterations have their priorities increased. In subsequent iterations, requests that were not previously scheduled are attempted to be satisfied first.

## Neighborhood Stochastic Search

The *Neighborhood Stochastic Search algorithm* (NSS) extends the request satisfaction variation of BD (Parjan and Chien 2023) to scale to large problem instances and enable scheduling with resource constraints. We present the pseudo-code in algorithm 1 and mention key sub-procedures. The first step for an agent is to compute, using GND, the sub-problem the agent is involved in solving. We denote this sub-problem as $\mathcal{N}$, which is itself a DCOP consisting of agents, $A(\mathcal{N}) \subseteq A$, and requests, $R(\mathcal{N}) \subseteq R$.

The procedure INITIALSOLUTION constructs an initial schedule for agent $a_i$. We consider two variations of this relying on fully decentralized algorithms:

---

**Algorithm 1: Neighborhood Stochastic Search for agent $a_i$**

**Input**: $H, A, R, S_{a_i}, D_{a_i}, C_{a_i}, \Upsilon_{a_i}$
**Output**: Schedule for agent $a_i$

1: $\mathcal{N}$ = COMPUTESUBPROBLEM($a_i, A, R, S_{a_i}, \Upsilon_{a_i}$)
2: sched = INITIALSOLUTION($\mathcal{N}, S_{a_i}, D_{a_i}, C_{a_i}$)
3: **while** not converged **do**
4:    com = MESSAGE($A(\mathcal{N})$, sched)
5:    **shuffle** $R(\mathcal{N})$
6:    **for** $r \in R(\mathcal{N})$ **do**
7:      isAssigned = STOCHASTICUPDATE($r$, sched, com)
8:      **if** isAssigned = TRUE **then**
9:        isScheduled = SCHEDULE($r$, sched, $S_{a_i}$)
10:      **end if**
11:      UPDATEDATASTRUCTS(isAssigned, isScheduled)
12:    **end for**
13: **end while**
14: **return** sched

---

1. *NSS-Random*. Agents construct random initial schedules, the typical initialization scheme for DSA.

2. *NSS-Decomposition*. The *Decomposition Heuristic* algorithm is used for the initial schedule.

The procedure MESSAGE encapsulates the communication between agents in a sub-problem. Each agent $a_i$ messages the subset of $R$ that it satisfied in the previous iteration to each agent in its sub-problem, $A(\mathcal{N})$, and receives the subsequent broadcast from those agents. The data structure com encapsulates these messages. The heuristic search is carried out in STOCHASTICUPDATE, which is adapted from the BD algorithm (Parjan and Chien 2023). This procedure updates the assignment of the agent and the request based on the communicated information. By assignment, we refer to whether or not the request *should* be scheduled by this agent. Let $m$ be the number of agents that satisfied request $r$ according to the broadcast. The assignment of a request $r$ is stochastically updated in the following ways.

- If agent $a_i$ is not assigned to $r$ and $r$ was not scheduled in the previous iteration, $a_i$ assigns to $r$.
- If agent $a_i$ is not assigned to $r$ and $r$ was scheduled in the previous iteration, $a_i$ remains unassigned to $r$.
- If agent $a_i$ is assigned to $r$ and $r$ was not scheduled in the previous iteration, $a_i$ will unassign with probability $P_u$.
- If agent $a_i$ is assigned to $r$ and $r$ was scheduled in the previous iteration, $a_i$ will unassign with probability $\frac{1}{m}$.

In the procedure SCHEDULE, if an assigned request fulfillment satisfies $C_{a_i}$ it is immediately inserted into the schedule. Otherwise, the scheduler can remove a conflicting request fulfillment from the schedule to free up resources. The removed request fulfillments are selected as the closest start time to the request fulfillment to insert. Allowing agents to de-schedule requests enables the algorithm to overcome getting stuck at local minima. Note that this algorithm relies on the parameter $P_u$. We use $P_u = 0.7$ as published by the authors of the BD algorithm (Parjan and Chien 2023).

The stochastic search performed in NSS mimics the search of BD with some key distinctions: the use of de-

composition to reduce size, de-allocating to overcome local-minima, the variation in initial schedule construction, and scheduling with resource constraints.

## Theoretical Analysis of Algorithms

We summarize the computational and communication complexity of the algorithms in the table below. Note that it is assumed that all agents have knowledge of the requests. Therefore, the fully decentralized algorithms incur no communication. To more exactly capture the complexity, we define $L$, the maximum size of a satellite's schedule. The value of $L$ is driven by the size of the horizon, $H$, the requests, $R$, and an agent's capabilities, $C_{a_i}$. We parameterize it to capture more exactly the complexity of our algorithms, but in practice $L << |R|$. Checking if a request fulfillment satisfies $C_{a_i}$ and inserting into a schedule are both $O(\log L)$ operations. We omit the $L$ factors in the computational complexity of NSS and SWO as it is subsumed by larger factors.

| Algorithm | Computation | Communication |
|---|---|---|
| Random | $O(|R| \log L)$ | N/A |
| Greedy Start Time | $O(|R| \log |R|)$ | N/A |
| Greedy Portfolio | $O(|R| \log |R|)$ | N/A |
| Decomposition Heuristic | $O(|R| + |R(\mathcal{N})| \log |R(\mathcal{N})|)$ | N/A |
| NSS | $O(|R| + k|R(\mathcal{N})||A(\mathcal{N})|)$ | $O(k|R(\mathcal{N})||A(\mathcal{N})|)$ |
| SWO | $O[k(|R|^2 + |R||A|)]$ | $O(L|A|)$ |

The communication and computation of the decentralized algorithms is shown per agent. The centralized algorithm, *Squeaky Wheel Optimization* (SWO), gives the complexity required of the centralized node. Centralized algorithms require communicating the final schedules to each agent, resulting in the $O(L \cdot |A|)$ communication cost. The NSS algorithm incurs a communication cost proportional to the size of the sub-problem. Here, $A(\mathcal{N})$ is the largest set of agents in a sub-problem and $R(\mathcal{N})$ is the largest set of requests in a sub-problem. For both NSS and SWO, we define $k$, the number of iterations of the algorithm.

The complexity analysis clearly shows that the fully decentralized algorithms are substantially more efficient than either NSS or SWO, and depending on the size of the sub-problems, NSS can be much more efficient than SWO, but incurs more communication. In comparison to MGM, DSA, or BD, NSS achieves a complexity parameterized by $|A(\mathcal{N})|$ and $|R(\mathcal{N})|$ in each iteration as opposed to $|A|$ and $|R|$.

## Experimental Setup and Results

The satellite constellation we simulate is modelled on a low Earth orbit Planet constellation (Planet 2023). There are 200 agents divided across 4 orbital planes. The constellation has two near sun-synchronous orbital planes at $95°$ inclinations composed of 95 satellites each. There are an additional two orbital planes at $52°$ inclinations with 5 satellites each. Each satellite has a single sensor that can slew to $60°$ off of nadir and an on-board memory capacity of 125 GB. Figure 5 shows the constellation centered around a sphere. We define the communication network topology such that each satellite messages the nearest agent in its neighborhood, and that agent must relay the message. This avoids satellites messaging without line-of-sight.

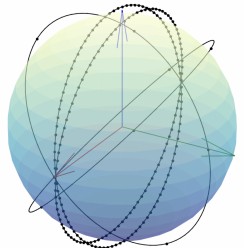

Figure 5: Visualization of the satellite constellation. Dots represent a satellite in an orbital plane.

We consider two ground stations for downlinks: the ASF Near Space Network Satellite Tracking Ground Station and the Guam Remote Ground Terminal System. A downlink is modelled as a constant bit stream of 62.5 MB/s for the duration of visibility of the ground stations.

The target set, $T$, is composed of 634 globally distributed ground targets (cities and volcanoes). We generate a campaign by selecting a random periodicity in the range $[4, 12]$. A periodicity of $n$ means each target is requested to be observed once within $n$ evenly spaced intervals during the scheduling horizon. For small problem instances, we reduce the periodicity to 2 and randomly remove requests to obtain a smaller set. The start of the scheduling horizon is randomly initialized during a week long simulation and the end of the horizon is fixed at 24 hours after the start time. We remove unsatisfiable requests based on satellite visibility during the horizon. The amount of memory required by a request fulfillment is sampled from a normal distribution with mean 50 MB and standard deviation 10 MB. The interval of time required to schedule a request fulfillment is fixed at 63 seconds (3 seconds for the observation and 30 seconds either side for slewing and processing).

Generating campaigns in this manner produces hard problem instances. By hard, we refer to the constraint graph structure discussed previously. Despite fixing the scheduling horizon at one day, it is the density of requests during the window (i.e. requests per epoch) that drives the difficulty of the scheduling problem. Large problem instances refer to problems with thousands of requests, resulting in millions of variables, whereas small problem instances contain less than 500 requests. We intend to release supplemental datasets containing problem instances when published.

## Results on Small Problem Instances

We compare the performance of the algorithms on small problem instances to an optimal solution, as well as the BD algorithm (Parjan and Chien 2023). We use a branch and bound search to obtain an optimal schedule for the constellation. The branch and bound algorithm can only execute on small problem instances due to computational constraints and the BD algorithm, likewise, due to communication constraints. We report the average gap in satisfaction to the optimal solution, the average execution time (per agent), and the average total messages exchanged over 50 randomly generated small problem instances in the table below.

| Algorithm | Average Gap to Optimal (%) | Average Execution Time (ms) | Average Messages |
|---|---|---|---|
| Random | 4.427 | < 1 | 0 |
| Greedy Start Time | 5.158 | < 1 | 0 |
| Greedy Portfolio | 3.807 | < 1 | 0 |
| Decomposition Heuristic | 2.271 | 1.42 | 0 |
| BD | 2.373 | 169.84 | 756,200 |
| NSS-Random | 0.580 | 43.24 | 66,690 |
| NSS-Decomposition | 0.409 | 39.66 | 63,180 |
| SWO | 0.012 | 2338.04 | < 500 |
| Branch and Bound | 0.0 | 6,670,695 | < 500 |

The results show that the centralized solution, SWO, achieves near-optimal performance. The NSS algorithms also achieve close to optimal performance, while the fully decentralized solutions are significantly further from the optimal solutions. Notably, the decomposition heuristic scheduling algorithm outperforms the other fully decentralized algorithms and the BD algorithm. In comparison to BD, the NSS algorithms achieve higher request satisfaction while possessing faster run-times and procuring an order of magnitude less messages. This supports the efficacy of GND in generating advantageous sub-problems and the theoretical analysis of the cost of NSS.

### Results on Large Problem Instances

We evaluate each scheduling algorithm against 100 randomly generated large problem instances. Note, solving a large problem instance optimally would likely take longer than the age of the universe. Figures 6 and 7 show the performance of the varying approaches. The horizontal lines in figure 6 represent the medians of the simulations. The NSS algorithms outperform the other decentralized solutions and are comparable in performance to the centralized approach, averaging just a %3 satisfaction decrease. In addition, the results enforce the effectiveness of the decomposition as the the fully decentralized approach outperforms the other baselines, and NSS-Decomposition slightly outperforms NSS-Random. Figure 7 also shows that as the density of requests grow, problem instances become more difficult. The constellation cannot satisfy all the requests, therefore coordinating to reduce redundancy of observations becomes crucial.

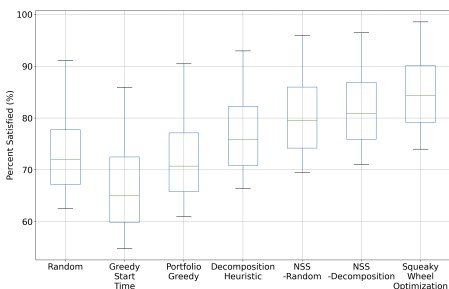

Figure 6: Spread of percentage satisfied requests over 100 large problem instances.

Figure 8 shows the execution time of the algorithms across problem instances. The simulations are executed in Java. Notice the non-linearity of the $y$-axis. The execution time of the decentralized approaches are reported as average time per agent. The fully decentralized algorithms all achieve a run-time between 1ms and 12ms. The NSS algorithms are an order of magnitude slower than the fully de-

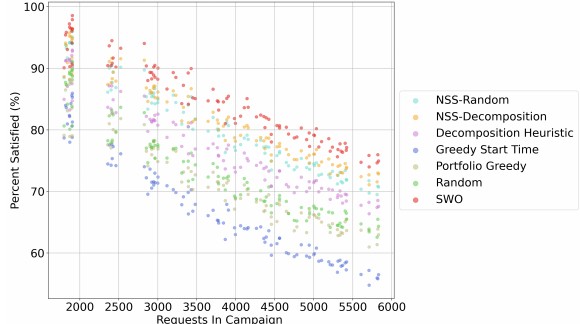

Figure 7: Satisfied requests (%) across 100 large problems

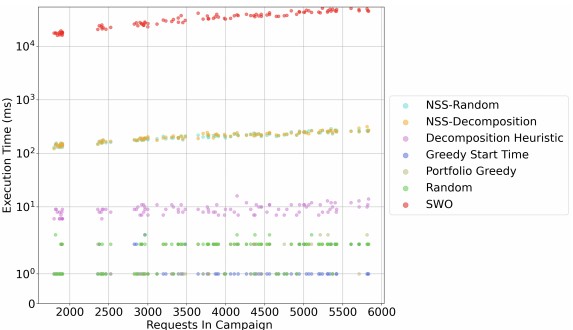

Figure 8: Execution time (ms) across 100 large problems.

centralized approaches. The centralized approach is another two orders of magnitude slower than the NSS algorithms. These results support the theoretical analysis.

While centralized algorithms will nearly always provide higher quality solutions, our GND-based solution is highly effective in the decentralized context, outperforming all the decentralized baselines. This demonstrates that high-quality schedules can be produced in very large-scale constellations by utilizing problem decomposition.

### Conclusion

A major barrier to applying existing DCOP algorithms to large-scale, real-world problems is their computation and communication complexities, specifically when dealing with highly connected constraint graphs. We propose a decomposition-based approach to the multi-satellite scheduling problem that is efficient in both time and message complexity and can scale to problems orders of magnitudes larger. Despite no quality guarantees, we have shown that solving well-constructed sub-problems can generate high quality global solutions while reducing the overall costs burdened by each agent. NSS can be adapted to other domains with different decomposition.

Beyond the application of scheduling a satellite constellation, many large multi-agent systems posses limiting constraints, and developing algorithms that work within those constraints is essential. Partitioning the global problem is one strategy to enable the broader application of DCOP solutions that have varying complexity.

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
