# OpenReview forum: "Decentralized, Decomposition-Based Observation Scheduling for a Large-Scale Satellite Constellation"
_icaps-conference.org/ICAPS/2024/Conference — ICAPS 2024_

### Official Review · Reviewer_UXFm · 2024-01-03

**Significance And Importance:** 2
**Soundness:** 4
**Novelty:** 3
**Clarity:** 4
**Overall Evaluation:** 2
**Confidence:** 3

**Weaknesses:**

1: Minor weaknesses that are easily fixable.

**Contributions Of The Paper:**

The paper proposes a distributed optimization algorithm for satellite constellation scheduling for earth observation tasks (COSP).
The highly constrained and connected global problem is deterministically and greedily partitioned into several smaller subproblems, such that satellites only need to coordinate their schedules for a subset of activities with other satellites in the same subgroup.
This allows for efficient computation of schedules that improve over the state of the art and come close to the quality of those produced by exact or heuristic centralized algorithms, while avoiding prohibitive computation or communication costs.

**Ethical Considerations:**

(1) Not Applicable: The paper does not have any ethical considerations to address

**Nomination For Best Paper:**

No

**Questions For Authors:**

Questions
----------
1) Does the model allow for different orbits along the same plane?
2) Is the COSP as described a new problem? If not, please add a reference in the appropriate section. If it is, what are the differences to prior literature?
3) Are request observation times contiguous intervals or arbitrary subsets of the scheduling period? (Same for request fulfillment intervals). Can a single satellite have multiple fulfillment opportunities for a single request?
4) Memory constraint: What happens if satellites are allowed to download data on later downlinks (and store it in between)?
5) The constraints do not enforce the requirement of "no task during downlink". Is this guaranteed by the input (i.e. no tasks exist that overlap downlink)?
6) How many iterations were used for SWO and NSS in the experimental evaluation?
7) line 459: How did you tune \rho? The resulting value 5 must be specific to the individual instance(s) used, I would suggest moving this result to the Experimental section.

**Reproducibility:**

3: Authors describe the implementation and domains in sufficient detail.

**Strengths Of The Paper:**

The paper is well written and easy to follow.
The COSP is both practically and theoretically relevant, while previously existing solution methods are clearly shown to be unsuitable for the scale of large constellations currently under deployment.
The problem formulation is complete and comprehensive.
The experimental evaluation is thorough and clearly shows the improvements over the state of the art as well as other baseline algorithms.

It is further positive that the authors intend to release the benchmark instance sets they generated to the public (releasing the source code as well would be a further bonus).

**Weaknesses Of The Paper:**

While the partitioning of requests provides good performance on the chosen benchmark instances, there is reason to believe that this may not generalize well to other instances, or problem variants:
- The inter-neighborhood delegation does not consider the number of requests already assigned to a plane - this might lead to e.g. some planes without any assigned requests, while other are completely overloaded. Consider e.g. the extreme case of two (nearly) parallel orbital planes, with satellites orbiting at slightly different heights. Those two planes provide supply to (roughly) the same requests, so according to the given delegation scheme, the one with the highest number of satellites will be assigned all of them, while the other one will remain empty.
- The description of the intra-neighborhood delegation is a bit hard to understand, but it seems to boil down to an arbitrary / random partitioning of requests into sub-planes. This works only if for each (or nearly each) request, at least one satellite of the appropriate "bias" (a better word might be "index" or "subgroup designation") will be in range within the request's time window. If request time windows become more narrow (such that less than \rho satellites are in range during the window), it gets increasingly more likely that none of the satellites from the subgroup assigned to the request is able to satisfy it at all. An intra-neighborhood delegation scheme that takes individual satellite and request timing information into account might fare better, but also require a different partition strategy since successive satellites are more similar with regards to their timing.

In the Theoretical Analysis, the comparison with the centralized algorithm is misleading. The given communication cost applies only to the central agent (which typically has arbitrarily high computation power and low computation cost compared to the others), all others have cost 0. Similar for computation costs. It would be better to list the maximum and average costs separately to allow for a fair comparison.

Minor issues:
- Problem formulation: Notation k(a_i) and m(a_i) might be confusing with function notation - why not use k_i or k_{a_i}? This notation is already in use for other parameters.
- Problem formulation: Difference between X and \mathcal{X} should be clarified
- Problem formulation, line 249: Set X^S is undefined
- The tables are hard to read due to font size. I propose using booktabs and floating tables across columns (table* environment)

Typos:
- line 429: "wider birth" -> "berth", even better "range"
- line 458: "the agent's" -> "the agents"
- line 672: "%3" -> "3%"
- line 673f: "as the the fully" -> duplicate "the"

Update after the response:
The fact that the "no request fulfillment during downlink" constraint is handled exclusively during the instance generation further makes the approach instance-specific, although this could be easily changed by adding appropriate constraints to the model.
For the final version, the authors should make it clear that their approach exploits characteristics of the specific instances (orbital configuration and request distribution) they evaluate it on and instances with other characteristics may require totally different partitioning schemes.

I would also suggest adding a reference to the Picard 2021 and/or Parjan and Chien 2023 papers to the problem definition, so that it is immediately clear where this specific COSP formulation originates from.

---

> ### Author Rebuttal · Authors · 2024-01-28
>
> Thank you for your valuable feedback and suggestions.
>
> In response to the questions:
> 1) Having multiple orbits within the same orbital plane is allowed in principle, but a constellation is unlikely to be designed this way. It restricts the global coverage of Earth. In any event, it would be reasonable to consider all the satellites in the orbital plane, even at different altitudes, as one neighborhood.
> 2) Many variations of COSP have been examined in the literature, and we discuss the ones that are closest to our problem.  We present a slight variation of COSP, which fits the DCOP definition, and we cite the papers that examine this variation.
> 3) The request intervals are contiguous intervals, which is typical for this application. An individual satellite may have zero, one or multiple request fulfillments per single request that all fall within the contiguous interval.
> 4) The memory constraint is dictated by the requirements of the application, although a variation in which this constraint is relaxed could be considered with our approach. It is relatively easy to handle situations in which a limited window of time is allowed for the downlink delay (such as one downlink). It is likely that longer delays would not be desirable for real time observation.
> 5) Yes, we enforce the “no request fulfillment during downlink” in the problem generation.
> 6) Both SWO and NSS had their maximum iterations set to 20, however they typically converged before that.
> 7) The value of rho was tuned using a grid search. Thanks for the suggestion to move the discussion of the value of rho to the experimental section.

---

### Official Review · Reviewer_X67W · 2024-01-21

**Significance And Importance:** 2
**Soundness:** 3
**Novelty:** 2
**Clarity:** 3
**Overall Evaluation:** 1
**Confidence:** 5

**Weaknesses:**

1: Minor weaknesses that are easily fixable.

**Contributions Of The Paper:**

The development of satellite constellations is nowaday important and is representing increasing possibilities for earth observation. Allocating observations to satellites, according to user requests, is known as a difficult problem, that has been studied for a long time. The paper proposes a distributed solving approach, known as Distributed Constrained Optimisation Problem (DCOP), that depends on observation windows, memory usage and downlink opportunities. A DCOP approach is interesting as it provides more reactivity to the constellation, can save communication, and can trade-off solving completness with the scale of the problem instances to address. However, the algorithm must save communications while solving the problem to be realistically deployed.The problem is formulated as a set of agent satelittes evolving in different orbital planes with opportunities on observation targets. The formulation is based on allocating an agent to observations with utility functions to maximise. The paper provides a decomposition approach to breackdown the main problem into sub problems, using two partitioning paradigms. The first makes use of inter-neighborhood delegation, where requests are delegated to other orbital planes depending on their properties. The second uses a type of periodic (cyclic) partitioning within a set of agents that belong to a same orbital plane. Authors propose a variations of domain heuristics and algorithms (decentralised, centralised and distributed) with communication complexity evaluations and performance caracterisation, evaluated over small and large problem instances. In particular a Neighborhood Stochastic Search (NSS), extending previous work (on broadcast decentralised), is highlighted by authors. The results are well presented and discussed, showing the interest of NSS to attain good solutions in reasonable time.

**Ethical Considerations:**

(1) Not Applicable: The paper does not have any ethical considerations to address

**Nomination For Best Paper:**

No

**Questions For Authors:**

How the communications between satellites could be performed? inter-satellite links (laser, radio) or others?
For your evaluations, problem instances are generated randomly, but how far they are from real data (downlink bandwidth, orbital periods LEO, latencies...)?
How the values on the utility functions can be interpreted in your results?

**Reproducibility:**

1: Difficult to reproduce because of missing detail.

**Strengths Of The Paper:**

The paper highlights a complex scheduling problem of a real world application.
The distributed approach is very interesting, given the current development of satellite constellations. This could be the focus of many future P&S applications.
Authors build on a comprehensive and solid state of the art, showing the innovation of the proposed approach.
The paper could create new perspectives for ICAPS, given the lack of distributed algorithms addressed in the P&S community.
The methodology also pertains the distributed computing (i.e. Edge) and Multi-Agent areas, where distributed protocols are elaborated with similar approaches.
The NSS method is also a promising approach for addressing this type of problem, according to the results.

**Weaknesses Of The Paper:**

Clearly, the decomposition method takes advantage of the geometric properties of the problems:
-having two different orbital planes involve different observation opportunities
-cyclic partitioning within an orbital plane can match the periodic observation opportunities of agent satellites.
The fact that the decomposition method is consistent with the problem structure is not discussed. This is an important point to mention, in spite of being part (even unconsciously) of the methodology.
The mathematical notation and the problem formulation are difficult to follow with several level of indexing. In particulat, the f() utility function seems to be semantic-free and is not really well presented. Concrete examples for f() would help the reader.
More generally, the paper lacks of concrete examples and illustrative data to help the reader.
The paper does not really exploit planning and scheduling background and contributes on distributed search techniques.
Other distributed approaches could be used (such as agreement protocols or argumentation techniques) but are not discussed.

---

> ### Author Rebuttal · Authors · 2024-01-28
>
> Thank you for your valuable feedback and suggestions.
>
> We will ensure to include more concrete examples and illustrative data to improve clarity.
>
> Space permitting, we will expand the discussion of other distributed approaches and whether they are appropriate, such as agreement protocols or argumentation techniques.
>
> Regarding the decomposition method, it takes advantage of the geometric properties of satellite constellations, which is what makes it so effective. However, the overall approach, which is based on a constraint graph decomposition grounded in geometric computation, can apply more broadly to other problems with different geometric characteristics.
>
> In response to the questions:
> 1) Modern inter-satellite communication is increasingly using lasers, however many satellites in orbit were manufactured with radio communications.
> 2) The simulated data is very close to what the real data would be. We simulate an LEO constellation that is based on an actual Planet constellation. We compute the overflight information using a physics based model where we propagate each satellite’s orbit over Earth. The downlink bandwidth, latencies, slewing time, image size, and other problem features are all selected based on reported values of in-orbit LEO satellites.
> 3) Each request satisfied is counted equally, and the utility function is the number of requests that were satisfied. For performance evaluation, we use the percentage of the requests satisfied (i.e., normalized utility between 0 and 1). It is possible to use more complex utility models, where the priority of requests contributes to the utility.

---

### Official Review · Reviewer_Xvmc · 2024-01-23

**Significance And Importance:** 2
**Soundness:** 3
**Novelty:** 2
**Clarity:** 3
**Confidence:** 4

**Weaknesses:**

0: Minor weaknesses requiring some work to be addressed for the paper to be accepted.

**Contributions Of The Paper:**

* The paper introduces a new decentralized algorithm called Neighborhood Stochastic Search (NSS) to maximize the number of requests satisfied for a multi-satellite observation scheduling problem. The algorithm is able to solve large-scale DCOPs.
* NSS exploits a problem decomposition provided by a method called the Geometric Neighborhood Decomposition heuristic (GND), and it uses restricted communications between the agents. In GND, subproblems are created by first allocating each request to a unique orbital plane based on geometric computations (so-called inter-neighborhood), and then by grouping some agents belong to the same orbital plane (so-called intra-neighborhood, based on the positions of the agents involved in the plane).
* Experiments are performed to compare two versions of NSS with other centralized and decentralized solving methods, both on small and large problems.

**Ethical Considerations:**

(1) Not Applicable: The paper does not have any ethical considerations to address

**Nomination For Best Paper:**

No

**Overall Evaluation:**

-1: (weak reject)

**Questions For Authors:**

(1) Could you deal with slewing times and temporally flexible observations?
(2) Please provide some features of the subproblems obtained.
(3) Did you use a MILP solver for the branch-and-bound method?
(4) How does the algorithm behave when the load must be shared? (see comments above)

**Reproducibility:**

2: Some details are missing, but the paper still appears to be replicable with some effort.

**Strengths Of The Paper:**

* This work concerns a practical application of planning techniques and uses realistic datasets.
* The related literature is mentioned and analyzed.
* The idea to partition the requests and the agents beforehand to decrease the computational complexity is interesting.
* Improvements in terms of solution quality and/or computation times are demonstrated with regards to several centralized and decentralized solution approaches, including the recent BD algorithm (Parjan and Chien 2023) that is also applicable to the problem considered here. The approach proposed also uses fewer communications when compared to other decentralized solutions.
* A theoretical analysis of candidate algorithms is provided, concerning computation times and communications.
* The very short computation times of NSS are compatible with an on-board usage.

**Weaknesses Of The Paper:**

* The downlink constraints are restrictive: there is a requirement to empty the memory during each downlink window, forbidding the possibility to wait for the next downlink window to download some remaining data. This restriction allows the authors to avoid propagating memory constraints over the whole horizon.

* The observation constraints are restrictive. More precisely, the time interval associated with an observation is assumed to be fixed beforehand (no temporal flexibility). This allows the authors to deal with selection constraints (constraints c_{s_j,s_l} on page 2) instead of scheduling (no-overlap) constraints over temporally flexible tasks. One issue is that the fixed observation interval h includes the slewing time, as mentioned at Line 211, but this slewing time should depend on the previous observation. This can have a strong impact in practice since the experiments consider a “pessimistic” 30 seconds either side for slewing and processing.

* The partitioning performed never considers the total load of each orbital plane or the load of each group of satellites within an orbital plane. For large-size instances, there may be so many requests that this is not an issue, since the satellites always have candidate observations, or in other words all groups of satellites are overloaded. But it is not clear how the algorithm behaves when the load must actually be shared or when observations have priorities or weights (more general than just maximizing the number of observations performed).

* Concerning the comparison with optimal methods, optimal schedules are obtained based on a branch and bound algorithm over the global problem. It is not clear why a MILP solver is not directly used for the small instances to try and get lower computation times. Also, in a MILP model, the observation constraints could be expressed in an aggregated form by enforcing constraints like sum_{s_j \in S} x(s_j) \leq 1 for sets of observations that pairwise overlap. Moreover, to better compare the relative contributions of GND and NSS, each subproblem produced by GND may be solved exactly using a MILP solver for the smallest instances.

* The features of the subproblems obtained in the experiments should be further detailed, such as the distribution of the number of candidate requests and satisfied requests in the subproblems, and the distribution of the number of candidate requests and satisfied requests between two downlink windows.

* There are some writing issues at some points:
   - in the definition of the DCOP, alpha is defined at Line 245 but used at Line 240
   - I do not see any difference between X_{a_i} and \mathcal{X}_{a_i}, since if we read the definition of alpha, X_{a_i} is the set of variables x such such x \in \mathcal{X}_{a_i}.
   - Line 249: why X^S and not X?
   - Writing O(2^{|A| |R|}) \simeq 10^{30102} is not sound.
   - The notion of distance between an orbital plane and a target is unclear: if the request window is large, is it meaningful to consider only the start, end and midpoints of window h(r)? Should the windows of the request fulfillment be considered instead?
   - Lines 490-491: “we also evaluate […] but” : the formulation could be improved.
   - Line 501: breaking ties with the exact supply : highest supply or lowest supply?
   - Line 545: what is m?
   - Algorithm 1: Line 11 is not detailed (function UpdateDataStructs).
   - First the bias is defined at Line 421, and then the authors write that there are two biases. The corresponding paragraph at Lines 433- 447 is unclear.

---

> ### Author Rebuttal · Authors · 2024-01-28
>
> Thank you for your valuable feedback and suggestions.
>
> The downlink model is dictated by the problem requirements, which does not allow delays. This is how most commercial satellites operate. We acknowledge this is an assumption and different applications that relax this will be considered in the future.
>
> Regarding weighted requests, the utility function can handle variable priorities, although we did not experiment with that. Both GND and NSS can be extended to more complex requests. We began to explore campaigns with varying priorities, for example NSS can factor the priority into the probability of assignment, and adjust the number of back-tracking steps.
>
> In response to the questions:
> 1) Yes, we can easily accommodate more complex slewing models via intra-agent constraints. Our approach is compatible with more complex slew models as GND is agnostic of intra-agent constraints. Temporally flexible scheduling is more challenging, but in any event, scheduling is largely dictated by the orbital constraints.
>
> 2) We will expand the description of subproblem features, e.g., for the large problem instances partitioned into 12 subproblems, the average of requests in a partition is 8.3% with variance 0.4% (nearly uniform). The mean of (total) requests per agent in a partition is 1.0% with variance .02%.
>
> 3) We did not use a MILP solver, however that would be another effective approach to obtain an optimal solution. The branch-and-bound algorithm is used only to provide an upper bound on the utility as a baseline to evaluate the performance of the various incomplete approaches, which are the focus of this paper. We do not believe that any complete method would be scalable to problem sizes we need to solve. But the suggestion to try a MILP solver is well taken.
>
> 4) Although the current utility model does not reward load balancing, GND can be modified to explicitly balance the load. The measures provided above suggest that in our experiments the heuristic did a good job of balancing requests across the partitions, without modification.
>
> Please note that m is defined on line 535, we break ties for SWO such that lower supply has higher priority, and the minimum distance to the orbital plane is used as a deterministic tie-breaking mechanism. Longer request windows would make this distance a worse heuristic of the supply, however it still functions as a tie-breaker.

---

### Meta-Review · Area_Chair_qwcm · 2024-02-05

**Recommendation:** Accept (Oral)
**Confidence:** 4

**Metareview:**

Positive points:
- Inspiring modeling as a distributed constraint optimization problem, with the help of the decomposition
- Neighborhood Stochastic Search algorithm is proposed, representing a promising approach
- Scalable approach demonstrated on large-scale instances
- Well-done presentation
- Valuable theoretical and experimental analysis

Negative points:
- The claim about the accommodation of more complex slewing models via intra-agent constraints is not true from the optimization point of view. So, there is still a gap between the proposed techniques and the actual application. Please reflect it in the paper's final version (see reviewer Xvmc's detailed comments).
- The proposed approach should be generalized. The reviewers recommend it as an important future work.

We further ask the authors to reflect on the reviews' comments in the final version of the paper in case of final acceptance.

**Ethical Considerations:**

(1) Not Applicable: The paper does not have any ethical considerations to address